# Optimal Subsidy Support for the Provision of Elderly Care Services in China Based on the Evolutionary Game Analysis

**DOI:** 10.3390/ijerph19052800

**Published:** 2022-02-28

**Authors:** Qiang Mu, Peng Guo, Ding Wang

**Affiliations:** School of Management, Northwestern Polytechnical University, Xi’an 710072, China; mq15npu@mail.nwpu.edu.cn (Q.M.); wangdingxibeigongda@mail.nwpu.edu.cn (D.W.)

**Keywords:** elderly care services, subsidy support, public–private partnership, evolutionary game

## Abstract

Public–private partnership is a type of cooperation that has been widely employed to alleviate contradictions between supply and demand in the elderly care industry in China. Based on evolutionary game theory, this paper mainly analyzes the effects of static subsidy and dynamic subsidy to private sectors and consumers on the evolution of the decision process for private investors, consumers, and government in the three-dimension system. The simulation results show that without active supervision, a higher subsidy to private sectors will not promote the provision of high-quality services when the cost saving is large. Furthermore, there exists a threshold value of the difference between the two types of subsidies such that elderly people will be encouraged to choose institutional care if the value exceeds the threshold. We also find that dynamic subsidy policy works more efficiently in promoting the provision of home-based care services.

## 1. Introduction

As more people live to an older age, population aging has moved into a priority place of the policy agenda of many countries including China. Data from the National Bureau of Statistics of China show that there are over 260 million individuals aged over 60, among which the number of disabled and semidisabled individuals is over 46 million, accounting for more than 18% of the total population in 2020 [1]. Furthermore, the decrease in the fertility rate in China in the past several years is leading to the shortage of labor force, which makes it more urgent to solve the problems caused by population aging to facilitate the sustainable development of society.

The rapidly increasing population of elderly people not only brings opportunities to the development of the elderly care industry, but also poses a great challenge to policy-makers in terms of the healthcare service provision as a result of the increasing demand for healthcare [2]. Shortage of elderly service provision, unprofessional service staff, and potential safety hazards may lead to poor quality of elderly care services, and thereby increase the risk to elderly people who often suffer from chronic diseases, disability, or semidisability [3]. Therefore, the sustainable provision of high-quality elderly care services is particularly important. To relieve the contradiction between the shortage of provision of high-quality elderly care services and the increasing demand for care services, public–private partnerships (PPP) pension projects have been widely appreciated and their application encouraged for developing elderly care services. Governments of developed countries such as France, Finland, and Germany have contracted out to private sectors to operate institutes of elderly care service [4]. The Chinese governments have also introduced substantial measures to support the sustainable development of public–private partnership (PPP) cooperation in elderly care services. However, in the operation of PPP projects in public areas, high costs from private sectors and substantial subsidies from governments are required. Incompletely identical objectives and asymmetric information, along with the incompleteness and externalities of contracts, enable private sectors to take opportunistic behaviors to maximize their own material payoffs [5,6,7]. For example, private sectors provide poor-quality elderly care services at high prices to pursue more benefits. Therefore, supervision and subsidy policies from governments are needed to formulate and regulate behaviors of participants in the elderly care service market, and thereby ensure the sustainable development of high-quality elderly care services. 

Regulatory decisions made by governments include subsidy strategy, rewarding private sectors that provide high-quality services and punishing those that take opportunistic behaviors. Sabry [8] pointed out that how and when to implement the supervision strategy or the subsidy strategy has a decisive influence on the success of public–private partnership cooperation. Extant studies about the provision of elderly care services are mainly focused on the motivation for launching PPP projects, the effect of the subsidy mechanism, and the management of risk factors that may impede project success [9,10,11]. Few studies concerning elderly care services have been conducted on how to subsidize services providers and the public to achieve project success. Meanwhile, it is generally acknowledged that the provision of elderly care services is associated with a set of independent institutes, organizations, and individuals (consumers), which can be treated as a complex system. The cooperation level between the participants in the provision of service determines whether high-quality elderly care services will be provided. Consequently, it is meaningful to incorporate major stakeholders and their dynamical process of decision making and interactions into the service provision model to accurately mimic reality. 

This paper analyzes the effect of subsidies to private sectors and consumers on the cooperation in the provision of elderly care services from the perspective of public–private partnerships. A multiagent evolutionary game model is established with the assumption that participants display bounded rationality. Evolutionary equilibriums of the complex system are obtained through theoretical analysis; the effects of different parameters on the evolution of strategies of participants under different scenarios are investigated. We also develop management suggestions for the governance of the provision of elderly care services. 

The remainder of the paper is structured as follows: Section 2 reviews the relevant literature. Section 3 introduces game relations among governments, private investors, and consumers, and describes model constructions and theoretical analysis. In Section 4, evolutionary dynamics of the model are simulated on Matlab (2017). Conclusions and suggestions are given in Section 5.

## 2. Literature Review

This study is associated with two streams of literature: the provision of elderly care service and evolutionary game theory.

### 2.1. Provision of Elderly CARE Services

Shortage of care beds and poor quality of elderly care services plagues governments and consumers. Using China as an example, China has about 30.53 long-term care beds per 1000 elderly individuals [12]. Moreover, the budget from government sectors for public long-term care is limited, which indicates that the resources governed by governments cannot meet the increasing demand for elderly care. In other words, private sector participation in the provision of elderly care services is indispensable and irreplaceable. 

Previous studies mainly focused on how governments formulate and regulate the provision of elderly care services and the quality and price competition of long-term care services. Inspired by real-world policies, Leporatti and Montefiori [13] examined elderly care programs and derived optimal policies to help policy-makers choose sustainable solutions to support home care. They found that in the context of information asymmetry, the implementation of the second-best outcome requires the level of care of the most subsidized households to be forced toward certain targets to avoid adverse selection. Valeria and Levantesi [14] proposed a de-risking strategy model for long-term care insurers. Their numerical application suggested that a de-risking strategy based on disability derivatives can be a viable solution to reduce the portfolio riskiness of long-term care insurers. Gori and Luppi [15] investigated how governments in six countries across Europe have regulated the delivery of cash-for-care schemes to dependent older people. By taking into account three analytical dimensions -- cash-for-care utilization dimension, professional support dimension, and the relationship between the delivery of cash-for-care and the delivery of the other publicly funded long-term care inputs—they showed that there has been a shared and increased interest in consolidating the regulation of cash-for-care delivery. Ma, Yi, and Hu [16] developed a punishment contract model and a revenue-sharing contract model to explore the strategic interactions between participants located in the supply chain for the provision of elderly care service. Results showed that increased dissatisfaction among the elderly reduces service quality and profits of providers and integrators, but increases the price of services; in the context of a punishment contract, increasing the punishment ratio has no effects on the supply chain of services, and in the context of a revenue-sharing contract, increasing the revenue-sharing ratio helps to improve the quality and price, but the overall profit of the supply chain decreases.

Researchers have also studied factors that affect the provision of elderly care services. Yakita [17] explored the economic development and long-term care provision by families, markets, and the state. The results show that the level of elderly care depends on the economic development stage. Moreover, public long-term care programs might be necessary to provide a minimum level of care for elderly people if the amount of elderly care provided by the family becomes too small. Bihan and Martin [18] examined the long-term care policy in France and argued that long-term care reform policy can only take place from a new compromise among three poles of protection: the family, the market, and the state.

Previous quantitative and qualitative studies analyzed the sustainable provision of elderly care services from perspectives of policy optimization, price, and quality competition. There are few studies concerning the strategic interaction between decision-makers associated with service provision. Yue and Lin [19] analyzed the effects of punishment and operating subsidy on the evolutionary stable strategies of private sectors and government regulators. They found that a higher level of subsidy can effectively improve the quality of services provided by private sectors, and increasing punishment can restrain investors from violating rules. He and Luo [20] developed an evolutionary game model to analyze the strategic behaviors of government regulators and private sectors in the provision of elderly care services. The model provides insights into how government regulators and private sectors make decisions under different scenarios. However, they analyzed the evolution of cooperation mainly between government regulators and private sectors, which does not take into account consumers that are indispensable participants. To improve the willingness of purchasing elderly care services, Xi’an city of China has proposed a subsidy mechanism to fund those who choose the institutional care. Additionally, the increasing population of elderly individuals makes the reputation effects yielded by consumers’ evaluation of the quality of services crucial factors that affect the project profit and public or private behavior, thereby having impacts on the development of elderly care services [21]. Consequently, the public should be included in the system of services provision system as independent decision-makers.

### 2.2. Evolutionary Game Theory and Its Application

The game–theoretic approach provides a quantitative framework to model, analyze, and predict behaviors of decision-makers in the interactions [22]. For instance, Debreu [23] included consumers as decision-makers in the economic system to study the equilibriums of the game model. As is known, orthodox game theory assumes that decision-makers are fully rational and can adopt the optimal strategy in each step. However, when it comes to multiagent interactions or group interactions in more complicated social systems, decision-makers do not always make the optimal strategy as they are bounded rationally [24]. Conversely, they may even make wrong decisions when playing games with others. Furthermore, the inability of orthodox game theory to demonstrate the process to reach the equilibriums makes it impossible to investigate the dynamical interactions between decision-makers. Evolutionary game theory (EGT) provides a distinct perspective to pinpoint the emergence and evolution of cooperation between agents and better reflects individuals’ behaviors as opposed to orthodox theory. 

Evolutionary game studies group behavior from the perspective of the evolutionary dynamic process. It holds that the knowledge concerning the cognitive and computational abilities of individuals is limited, and individuals have to adjust their strategies to learn from experiences and others’ strategies to determine the optimal decision. In our model, government regulators, private sectors, and consumers are participants in the complex system of long-term provision for elderly care services. Each decision-maker changes strategies over time to reach the Nash equilibrium: governments need to adapt the decision of supervision or subsidy to motivate private sectors to provide high-quality services, whereas private sectors have to adjust their strategies against government regulators and consumers to realize expected payoffs. Therefore, EGT is more suitable to investigate the provision process of elderly care services. 

EGT has been widely used to analyze social and economic problems such as R&D collaborations, poverty alleviation, and public goods provision. In [25], an evolutionary game model was used to analyze the behavioral strategies of agents in R&D collaborations, and game equilibrium was obtained to provide suggestions for collaborations between upstream and downstream R&D teams. Wan and Qie [26] explored the strategies of participants in the poverty alleviation ecosystem in China by using the EGT framework. Game equilibrium in the cooperation between the smart supply chain platform and governments was analyzed; the results show that enhancing the intelligence degree of the smart supply chain platform helps to transform the “blood transfusion” poverty alleviation to “hematopoietic” poverty alleviation, and decreases the dependence of poverty alleviation on government financial platform subsidies. Jiang and You [27] used EGT to investigate the strategic interactions of polluting enterprises (PEs), local government regulators (LG), and central government planners (CG) in China. They found that LGs will not insist on implementing environmental regulation policies negatively even without supervision from CG and thereby polluting firms will not always maintain the strategy of unlimited emission. These previous studies have shown that EGT is a powerful tool to explore and predict individuals’ behaviors in complex systems. 

## 3. Model

We now describe the situation under which governments, private investors, and consumers play games with each other to achieve their expected payoffs. Governments own the infrastructure necessary for providing elderly care services, such as unused apartments. Private sectors contract with governments to provide elderly care services in the pattern of public–private partnerships. Specifically, private sectors manage and operate the infrastructures to provide institutional and home-based care services, including daily care and medical care. Governments carry out policies such as subsidies for care beds, tax exemption, and supervision of service quality to promote the operation of elderly care institutions. For instance, in Beijing, private sector businesses operating elderly care institutes are eligible for tax subsidies, including business and income taxes. Consumers (elderly people) choose to purchase the services or not based on the quality provided in the market. 

Each of these decision-makers displays bounded rationality as their cognitive and computational abilities are limited; they all continuously adjust their strategies during the evolution. The model is an evolutionary game, as discussed in detail as follows.

### 3.1. Hypotheses and Descriptions

The following assumptions are proposed to facilitate the evolutionary game model. 

**A1.** *Three types of participants—private sectors, consumers, and government—participate in the provision of elderly care services. Private sectors can either provide high-quality services or poor-quality services, which are presented as* {HQ, LQ}*. Elderly people usually display different attitudes to institutional elderly care services. For instance, in the rural areas of Xi’an and Hohhot in China, elderly people with a low level of acceptance of elderly care services prefer to stay at home to have some elderly care services provided by their children. In other words, they are more willing to be nursed by their children than the nursing staff from institutions. In the urban area, elderly people show a higher acceptance of institutional care, which indicates that the elderly are more likely to live in institutions for elderly care services, if necessary. In our model, consumers refer to elderly people who are disabled or semidisabled, and their children cannot provide them with care services in time, thus daily care supplied by institutions is needed. It is worth noting that this assumption is derived from the situation that many young people in China cannot nurse their parents because of busy work schedules. Accordingly, the strategy set of consumers is denoted as*{IC,HC}*. IC presents the elderly who choose to have services in institutions—institutional care, and HC means that elderly people prefer to have care services at home, namely home-based care strategy. Governments can either actively or negatively supervise the quality of elderly care services provided by private sectors, and the strategy space of governments are “active supervision” and “negative supervision”, denoted as*{AS, NS}. 

**A2.** x(0<x<1)*is used to represent the probability that private sectors adopt the strategy of providing high-quality services, then private sectors provide poor-quality services with the probability*(1−x)*. Likewise,*y(0<y<1)*and*(1−y)*are employed to indicate the probabilities that consumers choose institutional care and home-based care, respectively. It is assumed that governments actively supervise or negatively supervise private sectors with the probability*z (0<z<1)*or*1−z. 

**A3.** *The unit costs of poor-quality and high-quality elderly care services provided at the price of*P*are*$C_{L}$*and*$C_{H}$*, respectively. Please note that*$C_{H}>C_{L}$*and*$\Delta C=C_{H}-C_{L}$. Q*measures the service demand in the elderly care market, denoted by the quantities of care beds in the model. Therefore,*$\left(P-C_{L}\right)Q$*and*$\left(P-C_{H}\right)Q$*indicate gains of private sectors yielded by the provision of poor-quality services and high-quality services. Additionally, there are reputation effects in the elderly care market. Specifically, the provision of high-quality services contributes to a higher reputation benefit*$r_{H}$*, while providing poor-quality services leads to a lower reputation benefit*$r_{L}$. 

**A4.** *Elderly people (consumers) who prefer institutional care pay*PQ*to have elderly care services and those who choose home-based care pay*PαQ*to have services at home.*α (0<α<1)*means the quantity of home-based care services provided is less than institutional care services. It is worth noting that some care providers will reduce costs of care services to improve net revenues to cover the costs of vacant beds if sales of services decrease. The parameter*γ (γ≥1)*is introduced to capture the phenomenon. To simplify the model,*γ*is set to*1/α*. Profits flow to the public yielded by PPP projects reflect the project performance, which is affected by private sectors’ efforts on service provision* [28]. *The level of service quality affects elderly people’s utility. Elderly people who choose institutional care will receive a high utility*
$E_{H}$
*(*$E_{H}>0$*) if private sectors provide high-quality nursing services, whereas poor-quality services lead to a low utility*
$E_{L}(E_{L}<0)$*. Note that the parameter*
β (0≤β≤1)
*is employed to scale the acceptance degree of elderly care services. Consequently, for consumers with a low-level acceptance of elderly care services, the utility obtained by having high-quality services is*
$\beta E_{H}$*, and*
$\left(1+\beta \right)E_{L}$
*for having poor-quality services.*


**A5.** *Governments pay a cost that is denoted as*C*for active supervision and the fine imposed on private sectors providing poor-quality services. Governments implement the incentive mechanism to promote the sustainable development of elderly care services. Governments subsidize private sectors according to the number of care beds. To mimic reality, we assume that when governments actively supervise the quality of services, private sectors will be subsidized*$QS_{1}$*to provide high-quality services, but will be subject to governmental punishment, which is measured by*F*if poor-quality services are provided. Meanwhile, reputation effects from consumers will bring additional social benefits to governments when active supervision is carried out and high-quality services are provided* [29]. $R_{H}$
*and*
$R_{L}$
*(*$R_{H}>R_{L}$*) measure the social benefits when consumers take strategies of institutional care and home-based care, respectively. We write*
$\Delta R=R_{H}-R_{L}$*. In addition, elderly people who purchase institutional care services are subsidized with*
$QS_{c}$*, those who choose home-based care services are subsidized with*
$\alpha QS_{c^{\prime }}$. $S_{c}$
*and*
$S_{c^{\prime }}$
*are subsidy factors.*


### 3.2. Construction of the Model 

According to the assumptions, the payoff matrix is presented in Table 1. Within each cell, private sectors’ payoff is in the first line, consumers’ payoff is in the second line, and the third line is the governments’ payoff. 

The equilibrium point of the evolutionary process can be obtained. According to the payoff matrix, the expected payoff of private sectors earned by choosing strategies of HQ and LQ are denoted as $I_{1}$ and $I_{2}$, respectively:$$I_{1}=y\left[Q(P-C_{H}\right)+QS_{1}+r_{H}]+\left(1-y\right)\left[Q(P-C_{H}\right)+QS_{1}+r_{L}]\;I_{2}=Q\left(P-C_{L}\right)-ZF+\left(1-Z\right)QS_{1}\;;$$

Assuming that the exponential Malthusian equation of growth in the probability of x is $N_{g}\left(t\right)=N_{0,g}e^{rg^{t}}$ [30], then private sectors’ replication dynamic equation can be derived as:(1)$$I\left(x\right)=\frac{dx}{dt}=x\left(1-x\right)[y\;(r_{H}-r_{L})+r_{L}-Q\Delta C+zQS_{1}+zF]$$

In the same way, the replication dynamic equations of consumers and governments are as follows:(2)$$C\left(y\right)=\frac{dy}{dt}=y\left(1-y\right)\left[x\left(1-\beta \right)E_{H}+x\beta E_{L}-\beta E_{L}-\alpha QS_{c^{\prime }}-\left(1-\alpha \right)QP+QS_{c}\right]\;$$
(3)$$G\left(z\right)=\frac{dz}{dt}=z\left(1-z\right)\left[xyR_{H}+x\left(1-y\right)R_{L}+\left(1-x\right)\left(F+QS_{1}\right)-C\right]\;$$

According to Equation (1), it can be derived that when $z=\frac{Q\Delta C-r_{L}-y\left(r_{H}-r_{L}\right)}{QS_{1}+F}$, I(x) equals 0, which means that the private sectors’ strategy is stable no matter which one it adopts. In the case of $z>\frac{Q\Delta C-r_{L}-y\left(r_{H}-r_{L}\right)}{QS_{1}+F}$, we can show that $\frac{dI\left(x\right)}{dt}=0$ when x=1, which represents that x=1 is a stable point, whereas x=0 is the equilibrium point when $z<\frac{Q\Delta C-r_{L}-y\left(r_{H}-r_{L}\right)}{QS_{1}+F}$. Consumers’ strategy remains stable when $x=\frac{\alpha QS_{c^{\prime }}+\beta E_{L}+\left(1-\alpha \right)QP+QS_{c}}{\left(1-\beta \right)E_{H}+\beta E_{L}}$. Similarly, y=1 is a stable point if $x>\frac{\alpha QS_{c^{\prime }}+\beta E_{L}+\left(1-\alpha \right)QP+QS_{c}}{\left(1-\beta \right)E_{H}+\beta E_{L}}$, whereas y=0 is the stable point when $x<\frac{\alpha QS_{c^{\prime }}+\beta E_{L}+\left(1-\alpha \right)QP+QS_{c}}{\left(1-\beta \right)E_{H}+\beta E_{L}}$. In the same way, if $y=\frac{C-xR_{L}-\left(1-x\right)\left(F+QS_{1}\right)}{x\left(R_{H}-R_{L}\right)}$, then G(z)=0, the governments’ strategy is stable. Active supervision (z=1) is the optimal strategy if $y>\frac{C-xR_{L}-\left(1-x\right)\left(F+QS_{1}\right)}{x\left(R_{H}-R_{L}\right)}$, otherwise negative supervision is the optimal strategy, namely z=0. 

The Jacobian matrix of the three-dimension dynamics system is
$$\begin{array}{c} J=\\ \left[\begin{array}{ccc} \left(1-2x\right)\left[y\left(r_{H}-r_{L}\right)+r_{L}-Q\Delta C+zQS_{1}+zF\right] & x\left(1-x\right)\left(r_{H}-r_{L}\right) & x\left(1-x\right)\left(QS_{1}+F\right)\\ \;y\left(1-y\right)\left[\left(1-\beta \right)E_{H}+\beta E_{L}\right] & \;\left(1-2y\right)\left[x\left(1-\beta \right)E_{H}+x\beta E_{L}-\beta E_{L}-\alpha QS_{c^{\prime }}-\left(1-\alpha \right)QP+QS_{c}\right] & 0\\ z\left(1-z\right)\left[yR_{H}+\left(1-y\right)R_{L}-\left(F+QS_{1}\right)\right] & z\left(1-z\right)x\Delta R & \left(1-2z\right)[xyR_{H}+x\left(1-y\right)R_{L}+\left(1-x\right)\left(F+QS_{1}\right) \end{array}\right] \end{array}$$

Based on the Jacobian matrix, we can conclude eight fixed points as equilibriums from G(x)=0, I(y)=0, and C(z)=0, namely $E_{1}=\left(0,0,0\right)$, $E_{2}=\left(0,0,1\right)$, $E_{3}=\left(0,1,0\right)$, $E_{4}=\left(1,0,0\right)$, $E_{5}=\left(0,1,1\right)$, $E_{6}=\left(1,0,1\right)$, $E_{7}=\left(1,1,0\right)$, $E_{8}=\left(1,1,1\right)$, and one mixed equilibrium $E_{9}=\left(x^{*},\;y^{*},z^{*}\right)$. According to Lyapunov stability [31], the equilibrium point is the evolutionary stable point of the system if eigenvalues of the Jacobian matrix are nonpositive numbers. For instance, the Jacobian matrix can be derived as follows in the case of $E_{1}=\left(0,0,0\right)$,
$$J_{1}=\left[\begin{array}{ccc} \lambda_{1} & & \\ & \lambda_{2} & \\ & & \lambda_{3} \end{array}\right]$$
where $\lambda_{1}=r_{L}-Q\Delta C$, $\lambda_{2}=-\beta E_{L}-\alpha QS_{c^{\prime }}-\left(1-\alpha \right)QP+QS_{c}$, and $\lambda_{3}=\left(F+QS_{1}\right)-C$. In the same way, the eigenvalues in accordance with equilibrium points are derived as follows in Table 2. Please note that eigenvalues yielded by $E_{9}=\left(x^{*},\;y^{*},z^{*}\right)$ are complex polynomials that are discussed in the simulation results. 

Based on the equilibria points and characteristic values, the stable points and evolutionary strategies can be analyzed. 

**Scenario 1.** $-\beta E_{L}-\alpha QS_{c^{\prime }}-\left(1-\alpha \right)QP+QS_{c}<0$

If $r_{H}-\left[Q\Delta C-QS_{1}-F\right]<0$ and $\left(F+QS_{1}\right)-C<0$, the equilibrium point $E_{1}=\left(0,0,0\right)$ is an ESS, which means that governments negatively supervise the quality of elderly care services, whereas private sectors provide poor-quality services and consumers remain with home-based care. However, $E_{2}=\left(0,0,1\right)$ is the evolutionarily stable point if $\left(F+QS_{1}\right)-C>0$. In other words, government sectors prefer to supervise elderly service actively when the sum of fine imposed on private sectors and subsidies are larger than the cost of active supervision. 

When $r_{L}-Q\Delta C+QS_{1}+F>0$ and $\left(1-\beta \right)E_{H}-\alpha QS_{c^{\prime }}-\left(1-\alpha \right)QP+QS_{c}>0$, $E_{8}=\left(1,1,1\right)$ is an ESS in the case of $R_{L}-C>0$, whereas $E_{7}=\left(1,1,0\right)$ is the stable point when $R_{H}-C<0$. This implies that private sectors will provide high-quality services when the payoff yielded by the reputation effect is larger than the sum of cost savings of providing low-quality services, subsidies, and fines imposed by governments. Meanwhile, governments will regulate the elderly care service market actively if the income generated by the reputation effect is higher than expenditures including supervision cost. However, in the case of $\left(1-\beta \right)E_{H}-\alpha QS_{c^{\prime }}-\left(1-\alpha \right)QP+QS_{c}<0$, $E_{6}=\left(1,0,1\right)$ and $E_{4}=\left(1,0,0\right)$ are evolutionary stable points when $R_{L}-C>0$ and $R_{L}-C<0$, respectively, which means that if the difference between utilities of consumers with different strategies is less than subsidies, consumers will not purchase elderly care services. 


**Scenario 2.**

$-\beta E_{L}-\alpha QS_{c^{\prime }}-\left(1-\alpha \right)QP+QS_{c}>0$



When $r_{H}-\left[Q\Delta C-QS_{1}-F\right]<0$ and $\left(F+QS_{1}\right)-C>0$, the equilibrium point $E_{5}=\left(0,1,1\right)$ is an ESS, whereas $E_{3}=\left(0,1,0\right)$ is an ESS if $\left(F+QS_{1}\right)-C<0$. The results show that the cost savings of poor-quality services prompt private sectors to provide consumers with low-quality elderly care services. 

When $r_{L}-Q\Delta C+QS_{1}+F>0$ and $\left(1-\beta \right)E_{H}-\alpha QS_{c^{\prime }}-\left(1-\alpha \right)QP+QS_{c}>0$, $E_{8}=\left(1,1,1\right)$ is an evolutionarily stable strategy if $R_{L}-C>0$; conversely, the equilibrium point $E_{7}=\left(1,1,0\right)$ is an ESS when $R_{H}-C<0$. Similar to case b in (1), we can show that $E_{6}=\left(1,0,1\right)$ and $E_{4}=\left(1,0,0\right)$ are evolutionary stable strategies when $R_{L}-C>0$ and $R_{L}-C<0$, respectively, if $\left(1-\beta \right)E_{H}-\alpha QS_{c^{\prime }}-\left(1-\alpha \right)QP+QS_{c}<0$. Based on the analysis of (1) and (2), the invisible reputation effects and the difference between high-quality services and poor-quality services, subsidies, and fines carried out by governments are critical factors of the promotion of high-quality service provision. Consumers tend to purchase elderly care services when the difference between utilities of two types of attitudes toward elderly care services is larger than the differences of subsidies from governments.

## 4. Simulation Results

Based on the theoretical analysis under different situations, we now use the simulations performed on Matlab2017 to investigate the effects of punishment, subsidies to private sectors, and consumers on the provision of elderly service. We assume that initial states of private sectors, government, and consumers are equal to 0.5. The parameters are initialized as displayed in Figure 1, Figure 2, Figure 3, Figure 4, Figure 5, Figure 6, Figure 7, Figure 8, Figure 9, Figure 10 and Figure 11. 

### 4.1. The Effect of Static Subsidy from Governments 

Figure 1 presents the evolutionary paths of the multiagent system under different subsidies from governments to private sectors. $S_{c^{\prime }}$ and $S_{c}$ are set to 5 to avoid the interference of subsidies to consumers. It is evident that the evolution trends of strategies of private sectors and governments are similar and they are in a cyclical state, which indicates that there is no stable evolution strategy. When governments do not subsidize private sectors, namely $S_{1}=0$, governments will carry out active supervision with a relatively high probability, and there is a higher likelihood that private sectors provide high-quality services. Additionally, consumers will choose the strategy of institutional care with a higher probability, however, as the scale of subsidy to private sectors increases, the probability declines. Moreover, a closer observation of Figure 1 finds that the circular path expands as the subsidy increases when $S_{1}>0$. This occurs because a low payoff yielded by reputation effect and the punishment imposed on private sectors cannot promote governments to actively supervise the quality of services provided by private sectors sustainably. Consequently, private sectors choose the HQ strategy when active supervision is implemented but will serve consumers with poor-quality services when negative supervision is carried out. The quality level of elderly care services decreases as governments expand the scale of subsidy to private sectors without continuously active supervision, and therefore consumers are more likely to adopt home-based care for elderly care services. 

Figure 2 shows that when $S_{1}=0$, the probability for private sectors to provide high-quality services decreases first and then increases. This occurs because governments actively supervise the quality with a relatively high and stable probability, as depicted in Figure 4. In cases of $S_{1}>0$, the probability of providing high-quality elderly care services increases with the increase of the subsidy scale. However, the evolution path fluctuates over time and no stable strategy exists, which verifies the result shown in Figure 1. Furthermore, as the scale of subsidy increases, governments are more likely to adopt negative supervision because of excessive expenditure cost, and the evolution path fluctuates more dramatically, as shown in Figure 4. Figure 3 shows that consumers maintain the strategy of HC under any given values of $S_{1}$. 

To investigate the mechanism that promotes the provision of high-quality elderly care services, the effect of the penalty is explored and presented in Figure 5. The increasing scale of the penalty to poor-quality services increases the probability of active supervision and therefore motivates private sectors to provide high-quality services with a relatively higher probability. However, more fines cannot induce private sectors to produce high-quality services continuously and stably. 

The subsidy to elderly care people also affects the evolution of elderly service provision [32]. Figure 6 presents the evolution of strategies under different subsidies to consumers. $S_{c}>S_{c^{\prime }}$ indicates that the option of institutional care is widely employed, while $S_{c}<S_{c^{\prime }}$ means home-based care strategy is encouraged. It can be seen in Figure 6 that when the subsidy to elderly people who prefer institutional care is significantly larger than that to those choosing home-based care ($S_{c}=10,S_{c^{\prime }}=5$ and $S_{c}=10,S_{c^{\prime }}=0$), the strategy of the elderly evolves to IC (institutional care). At the same time, strategies of private sectors and governments evolve into cyclical states. Furthermore, the three-dimensional system eventually evolves into a circular state with decreasing $S_{c}$. When $S_{c}<S_{c^{\prime }},$ the evolution of strategies of private sectors and governments are in cyclical states, consumers choose the institutional care strategy with a certain probability, and thereby there exists no ESS. The results reveal that there exists a threshold value of $S_{c}-S_{c^{\prime }}$ such that elderly people will be encouraged to choose institutional care if $S_{c}-S_{c^{\prime }}$ exceeds the threshold. 

Elderly care services produced by private sectors develop rapidly to meet the increasing demand for nursing homes and institutions. How to regulate and improve the developmental sustainability of the elderly care service market has been a great challenge for governments. The government of Henan province in China decided to include the quality of the elderly care market into the “Local Government Performance Evaluation System” to promote the provision and consumption of elderly care services. Local governments will gain extra reputation and social benefits from superior governments and consumers if policy-makers make appropriate decisions to induce the provision of high-quality services. Accordingly, we proceed to perform the simulation with higher values of $R_{H}$ and $R_{L}$. 

Figure 7 depicts the evolution of strategies of participants in the service provision system under different cost savings of providing poor-quality services. In the case where the difference between the costs of high-quality services and poor-quality services is large (Figure 7a), a smaller subsidy (such as $S_{1}=0$ and $S_{1}=2$) to private sectors leads the evolution system to the ESS $E_{9}=\left(x^{*},\;y^{*},z^{*}\right)$, where governments supervise the quality of care services with a certain probability. Simultaneously, private sectors provide high-quality services with the probability $x^{*}$($0<x^{*}<1$), and elderly people purchase institutional care with the probability $y^{*}(0<y^{*}<1)$. Furthermore, when $S_{1}$ increases to 4, we can obtain the ESS $E_{6}=\left(1,0,1\right)$, where elderly people retain the strategy of home-based care. However, the evolution system eventually evolves to the cyclical state as $S_{1}$ increases, and a larger subsidy leads to a lower probability of high-quality services provision. Therefore, the willingness of consumers to adopt the strategy of institutional care decreased. In the case where the difference between the costs is a medium size, as shown in Figure 7b, $E_{6}=\left(1,0,1\right)$ is the ESS when $S_{1}<4$. The strategy of consumers evolves to HC, on the contrary, governments tend to implement the active supervision strategy due to the reputation effect of the public, which is similar to the case in Figure 7c. The circular state is the final state when $S_{1}>4$. Figure 7c shows that when the cost saving of providing poor quality services is small, $E_{4}=\left(1,0,0\right)$ is the ESS of the evolution system. Moreover, the growth rate of evolution of private sectors to providing high-quality services decreases as the subsidy from government increases. Specifically, the growth rate of the evolution of private sectors drops dramatically when the subsidy is more than the cost of poor-quality services. In addition, a comparison of Figure 7a–c shows that private sectors will produce high-quality services even if the subsidy is less than $C_{L}$ when the difference between costs of high-quality and poor-quality services is medium or small size, as opposed to large size. This occurs because the reputation effect from the public, the subsidy, and the penalty from governments exceeds the cost-saving of providing poor-quality services, as proved in case (1) in Section 3.2. 

Figure 8 describes the evolution of strategies under different subsidies to elderly people. 

As shown in findings unfolded in Figure 6, identical subsidies to elderly people choosing institutional care and home-based care service lead the evolution system to the cyclical state in cases of ΔC=6 and ΔC=9, while (1, 0, 0) becomes the ESS when ΔC=3. When the subsidy to elderly people preferring institutional care is significantly larger than that to those who choose home-based care strategy, $E_{8}=\left(1,1,1\right)$ is the ESS, which indicates that governments and private sectors adopt strategies of AS and HQ, respectively. It can also be seen that the evolution rate increases as ΔC decreases. Comparing with the results shown in Figure 6, we can conclude that a larger amount of subsidy to the purchase of institutional care services and higher reputation benefits can promote the emergence of $E_{8}=\left(1,1,1\right)$. However, the equilibrium vanishes and strategies evolve into a cyclical state when $S_{c}<10$, as indicated in Figure 8a,b. $E_{4}=\left(1,0,0\right)$ is the ESS when ΔC=3 in cases of $S_{c}<10$, implying that a small difference of service cost promotes private sectors to provide high-quality services.

Figure 9 is plotted to investigate the effect of the acceptance level of elderly care services of consumers on strategies evolution. It is evident that subsidizing elderly people who hold institutional care strategy induces the emergence of an ESS: $E_{8}=\left(1,1,1\right)$ when β<1, in Figure 9a. However, the system does not have an evolutionary stable point when $S_{c}\leq S_{c^{\prime }}$, as presented in Figure 9b,c. 

### 4.2. The Effect of Dynamic Subsidy from Governments

Either a static subsidy policy or a dynamic subsidy policy is carried out to foster an industry that is in its infancy. In this section, we examine the effect of dynamic subsidy policy instead of the static subsidy policy. We employ $S_{1d}\left(1-x\right)$ to represent the subsidy to private sectors, where $S_{1d}$ denotes the maximum value of the subsidy. $S_{cd}\left(1-y\right)$ is the subsidy to consumers who choose institutional care, and $S_{cd}$ measures the maximum value of subsidy from governments [20]. In the same way, $S_{c^{\prime }d}y$ indicates the subsidy to consumers who choose home-based care, and $S_{c^{\prime }d}$ scales the maximum value of the subsidy. 

As was done in Section 3.2, the stable points and evolutionary strategies can be analyzed as follows:

**Scenario 1**. $-\beta E_{L}-\alpha QS_{c^{\prime }d}y-\left(1-\alpha \right)QP+\left(1-y\right)QS_{cd}<0$

If $r_{H}-\left[Q\Delta C-QS_{1d}-F\right]<0$ and $\left(F+QS_{1d}\right)-C<0$, the equilibrium point $E_{1}=\left(0,0,0\right)$ is an ESS, and $E_{2}=\left(0,0,1\right)$ is the evolutionarily stable point if $\left(F+QS_{1d}\left(1-x\right)\right)-C>0$. 

When $r_{L}-Q\Delta C+QS_{1d}+F>0$ and $\left(1-\beta \right)E_{H}-\alpha QS_{c^{\prime }d}-\left(1-\alpha \right)QP+QS_{cd}>0$, $E_{8}=\left(1,1,1\right)$ is an ESS in the case of $R_{L}-C>0$, whereas $E_{7}=\left(1,1,0\right)$ is the stable point when $R_{H}-C<0$. However, in the case of
$\left(1-\beta \right)E_{H}-\alpha QS_{c^{\prime }d}-\left(1-\alpha \right)QP+QS_{cd}<0$
, $E_{6}=\left(1,0,1\right)$ and $E_{4}=\left(1,0,0\right)$ are evolutionary stable points when $R_{L}-C>0$ and $R_{L}-C<0$, respectively, which means that if the difference between utilities of consumers taking different strategies is less than subsidies, then consumers will not purchase elderly care services. 

**Scenario 2**. $-\beta E_{L}-\alpha QS_{c^{\prime }d}y-\left(1-\alpha \right)QP+\left(1-y\right)QS_{cd}>0$

When $r_{H}-\left[Q\Delta C-QS_{1d}-F\right]<0$ and $\left(F+QS_{1d}\right)-C>0$, the equilibrium point $E_{5}=\left(0,1,1\right)$ is an ESS, whereas $E_{3}=\left(0,1,0\right)$ is an ESS if $\left(F+QS_{1d}\right)-C<0$. The results show that the cost savings of poor-quality services prompt private sectors to provide consumers with low-quality elderly care services. 

When $r_{L}-Q\Delta C+QS_{1d}+F>0$ and $\left(1-\beta \right)E_{H}-\alpha QS_{c^{\prime }d}-\left(1-\alpha \right)QP+QS_{cd}>0$, $E_{8}=\left(1,1,1\right)$ is an evolutionarily stable strategy if $R_{L}-C>0$, conversely, the equilibrium point $E_{7}=\left(1,1,0\right)$ is an ESS when $R_{H}-C<0$. Similarly to case b in (1), we can obtain that $E_{6}=\left(1,0,1\right)$ and $E_{4}=\left(1,0,0\right)$ are evolutionary stable strategies when $R_{L}-C>0$ and $R_{L}-C<0$ respectively if $\left(1-\beta \right)E_{H}-\alpha QS_{c^{\prime }d}-\left(1-\alpha \right)QP+QS_{cd}<0$. 

Simulations have been conducted to facilitate the understanding of the analysis above. Figure 10 presents the evolution of strategies of private sectors, consumers, and governments under dynamic subsidy to private sectors and static subsidy to consumers. $E_{8}=\left(1,1,1\right)$ is the ESS when $S_{cd}=10$, $S_{c^{\prime }d}=$5, while $E_{6}=\left(1,0,1\right)$ becomes the ESS when $S_{cd}\leq S_{c^{\prime }d}$. From a comparison of Figure 8b and Figure 10a, we can find that the dynamic subsidy policy accelerates the evolution rate of consumers to adopt institutional care but slows the evolution rate of private sectors to providing high-quality services when $S_{\mathrm{cd}}=10$, and $S_{c^{\prime }d}=5$. Closer examination of Figure 10b,c shows that governments are more likely to take active supervision when $S_{c^{\prime }d}>S_{cd}$. Moreover, when comparing Figure 10b with Figure 7b, we observe that strategies of governments evolve more quickly to active supervision and therefore $E_{6}=\left(1,0,1\right)$ becomes the ESS as opposed to the cyclical state when a static subsidy is implemented. Comparing among Figure 7b, Figure 8b, and Figure 10 shows that dynamic subsidy to private sectors contributes more to the ESS $E_{6}=\left(1,0,1\right)$. In addition, the evolution rate to ESS declines with the increasing maximum value of the dynamic subsidy to private sectors. 

To further test how dynamic subsidy policy affects the strategy evolution, we assume that governments subsidize private sectors and consumers dynamically at the same time. As shown in Figure 11a, the evolutionarily stable strategy is (1, $y^{*'}$, 1), where $0<y^{*'}<1$. That is, the elderly people choose institutional care with a certain probability $y^{*'}$ regardless of the maximum subsidy $S_{1d}$ when the maximum value of the subsidy to consumers taking the strategy of institutional care is more than that to those who choose home-based care. Comparison between Figure 10b,c and Figure 11 shows that the dynamic subsidy to consumers decreases the evolution rate of the system to the equilibrium point $E_{6}=\left(1,0,1\right)$. 

## 5. Conclusions and Suggestions

Based on the bounded rationality of participants, this study develops an evolutionary game model among private sectors, consumers, and governments to investigate the problem of the provision of quality services in the elderly care service industry using an agent-based computational approach. Specifically, we assume that governments carry out two types of subsidy policy—the static subsidy and the dynamic subsidy—and the evolution of strategies of participants related to the service provision system under different subsidy policies is explored. The stability analysis demonstrated that the cost difference between high-quality services and poor-quality services, the reputation effect from the public, and scales of subsidy all play important roles in the sustainable provision of high-quality elderly care services. The simulation results show that: Private sectors provide high-quality services when active supervision is implemented but serve consumers with poor-quality services when negative supervision is carried out even if governments implement a relatively higher subsidy. Larger fines imposed on private sectors increases the probability of active supervision and therefore motivate private sectors to provide high-quality services with a relatively higher probability, yet cannot induce private sectors to keep providing high-quality services stably and continuously. Furthermore, there exists a threshold value of $S_{c}-S_{c^{\prime }}$ such that elderly people will be encouraged to choose institutional care if $S_{c}-S_{c^{\prime }}$ exceeds the threshold. Private sectors will produce high-quality services even if the subsidy is less than the cost of low-quality services when the difference between costs of high-quality and poor-quality services is small, but not when it is large. Moreover, when the difference between costs of high-quality and poor-quality services is of medium or large size, a larger amount of subsidy to the purchase of institutional care services and higher reputation benefits from the public can promote the emergence of an equilibrium point (1, 1, 1). However, the equilibrium vanishes and strategies evolve into a cyclical state when $S_{c}$ decreases. Finally, the dynamic subsidy to private sectors accelerates the evolution rate of consumers to adopt home-based care but slows the evolution rate of private sectors to provide high-quality services. The evolution rate to ESS declines with the increasing maximum value of the dynamic subsidy to private sectors. 

To promote the sustainable provision of elderly care services and the development of the elderly care industry, the following specific measures are put forward according to the analysis above. Firstly, a friendly environment that nurses the elderly care service institution should be constructed and optimized. Subsidizing the operation cost and training and providing senior nursing talents would reduce the difference between costs of providing high-quality services and low-quality services. Secondly, the policy system that financially supports the development of the elderly care industry should be optimized. Appropriate subsidy scale to private sectors and a high level of probability of active supervision need to be carried out to prevent investors from breaching the contract when the industry is in its infancy. In addition, high-quality home-based care services can be promoted by a dynamic subsidy mechanism. Thirdly, the reputation effect can be included in the service evaluation mechanism to promote the provision of high-quality services.

## Figures and Tables

**Figure 1 ijerph-19-02800-f001:**
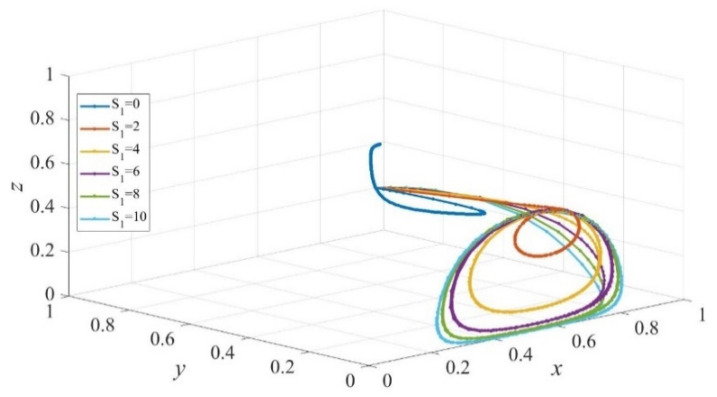
The evolution path of strategies of private sectors, governments, and consumers under different scales of subsidy to private sectors. Parameters: Q=50, P=18, ΔC=6, $S_{c}=5,$ $S_{c^{\prime }}=5$, $E_{H}=200$, $E_{L}=-50$, β=0.8, $r_{H}=260$, $r_{L}=220,R_{H}=150$, $R_{L}=100$, C=80, F=85, α=0.6.

**Figure 2 ijerph-19-02800-f002:**
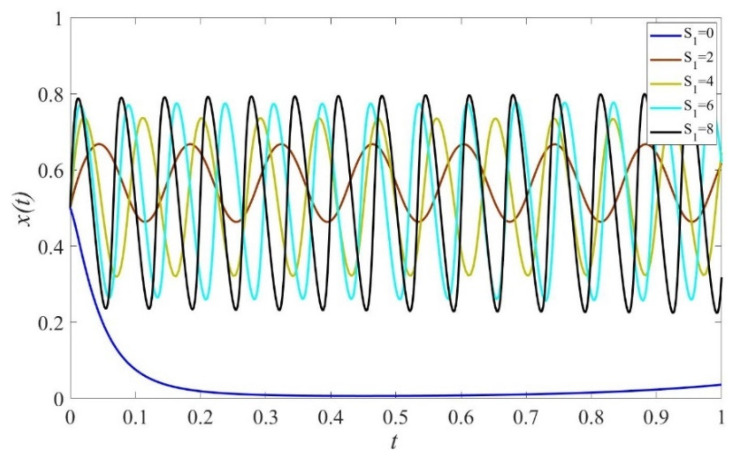
The evolution path of strategies of private sectors under different scales of subsidy. Parameters: Q=50, P=18, ΔC=6, $S_{c}=5,$ $S_{c^{\prime }}=5$, $E_{H}=200$, $E_{L}=-50$, β=0.8, $r_{H}=260$, $r_{L}=220,R_{H}=150$, $R_{L}=100$, C=80, F=85, α=0.6.

**Figure 3 ijerph-19-02800-f003:**
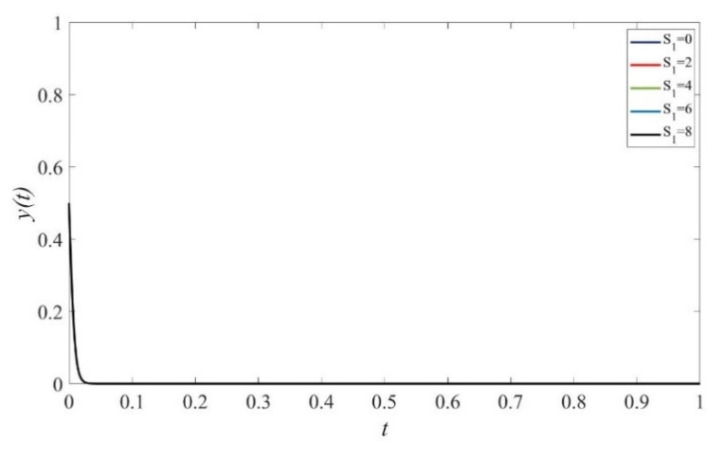
The evolution path of strategies of consumers under different scales of subsidy. Parameters: Q=50, P=18, ΔC=6, $S_{c}=5,$ $S_{c^{\prime }}=5$, $E_{H}=200$, $E_{L}=-50$, β=0.8, $r_{H}=260$, $r_{L}=220,R_{H}=150$, $R_{L}=100$, C=80, F=85, α=0.6.

**Figure 4 ijerph-19-02800-f004:**
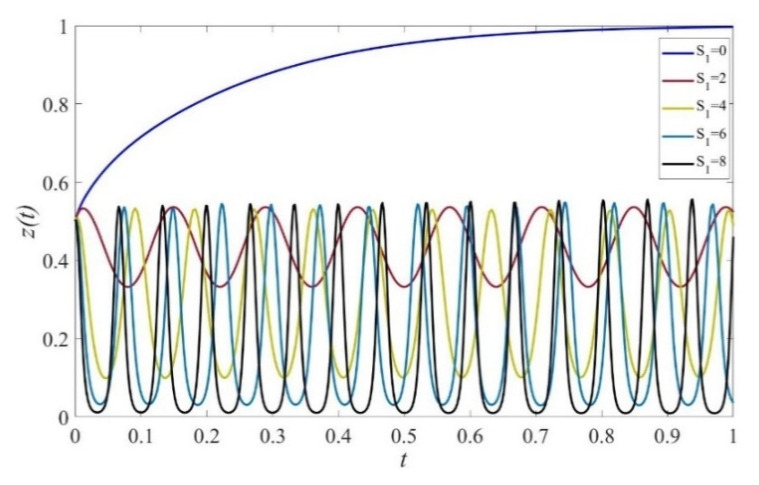
The evolution path of the strategy of governments under different scales of subsidy. Parameters: Q=50, P=18, ΔC=6, $S_{c}=5,$ $S_{c^{\prime }}=5$, $E_{H}=200$, $E_{L}=-50$, β=0.8, $r_{H}=260$, $r_{L}=220,R_{H}=150$, $R_{L}=100$, C=80, F=85, α=0.6.

**Figure 5 ijerph-19-02800-f005:**
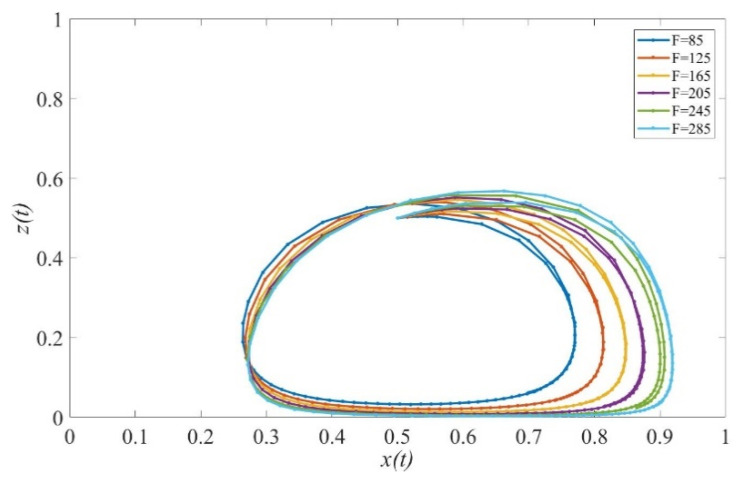
The coevolution of strategies of private sectors and governments under different penalties. Parameters: Q=50, P=18, ΔC= 6, $S_{c}=5,$ $S_{c^{\prime }}=5$, $S_{1}=6$, $E_{H}=200$, $E_{L}=-50$, β=0.8, $r_{H}=260$, $r_{L}=220,R_{H}=150$, $R_{L}=100$, C=80, α=0.6.

**Figure 6 ijerph-19-02800-f006:**
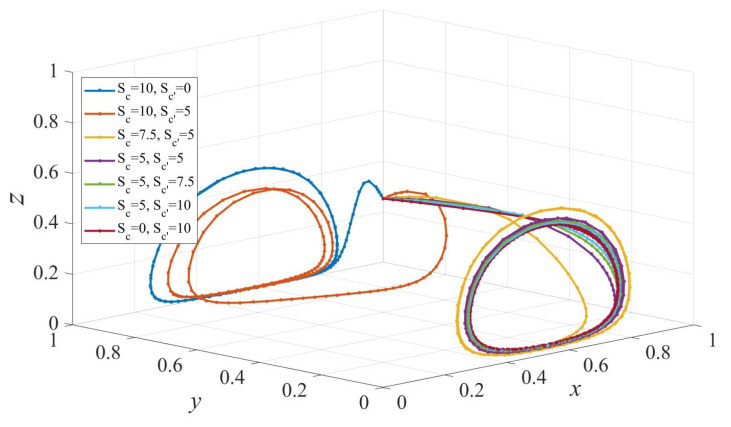
The evolution path of strategies of governments, private sectors and consumers under different scales of subsidy to consumers. Parameters: Q=50, P=18, ΔC= 6, $S_{1}=6$, $E_{H}=200$, $E_{L}=-50$, β=0.8, $r_{H}=260$, $r_{L}=220,R_{H}=150$, $R_{L}=100$, C=80, F=85, α=0.6.

**Figure 7 ijerph-19-02800-f007:**
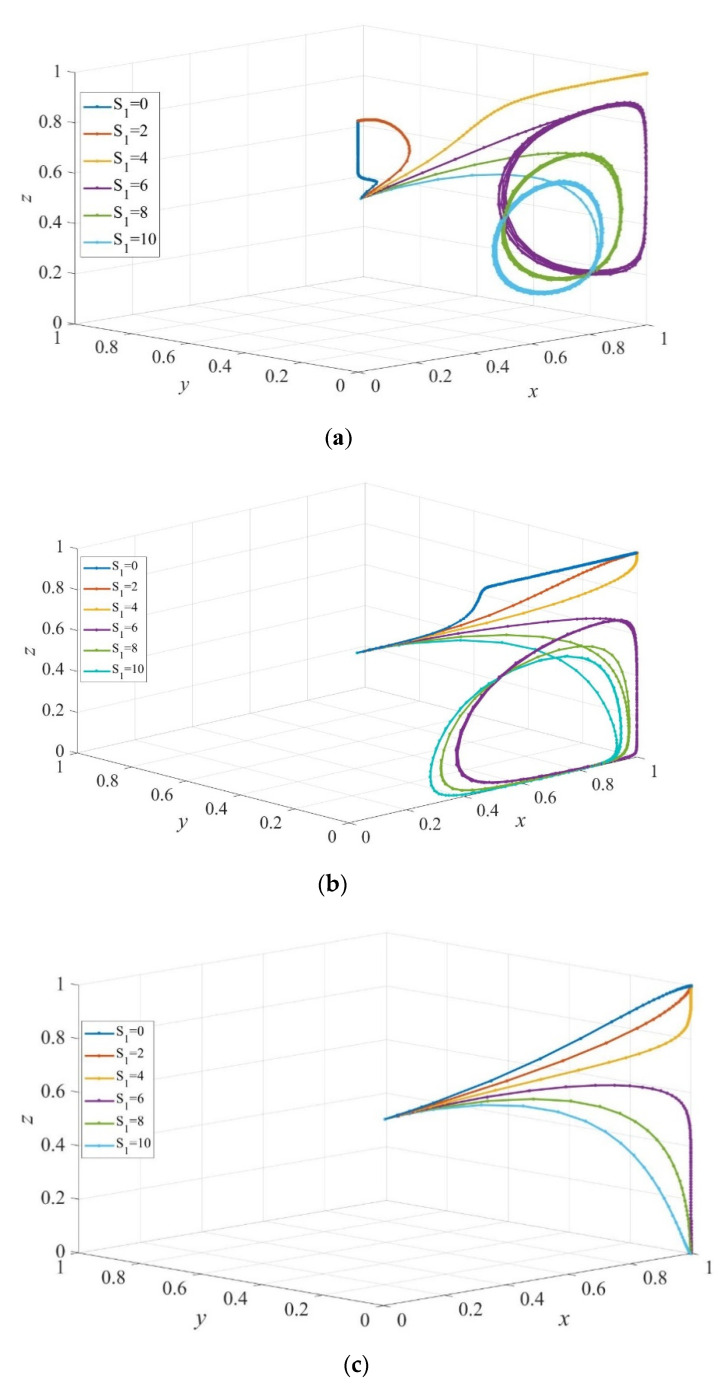
The evolution path of strategies of private sectors, consumers, and governments under different cost savings. ΔC = 9 in (**a**), ΔC= 6 in (**b**), and ΔC= 3 in (**c**). Parameters: Q=50, P=18, $S_{c}=S_{c^{\prime }}=5$, $E_{H}=200$, $E_{L}=-50$, β=0.8, $r_{H}=260$, $r_{L}=220,\;R_{H}=390$, $R_{L}=340$, C=80, F=85, α=0.6.

**Figure 8 ijerph-19-02800-f008:**
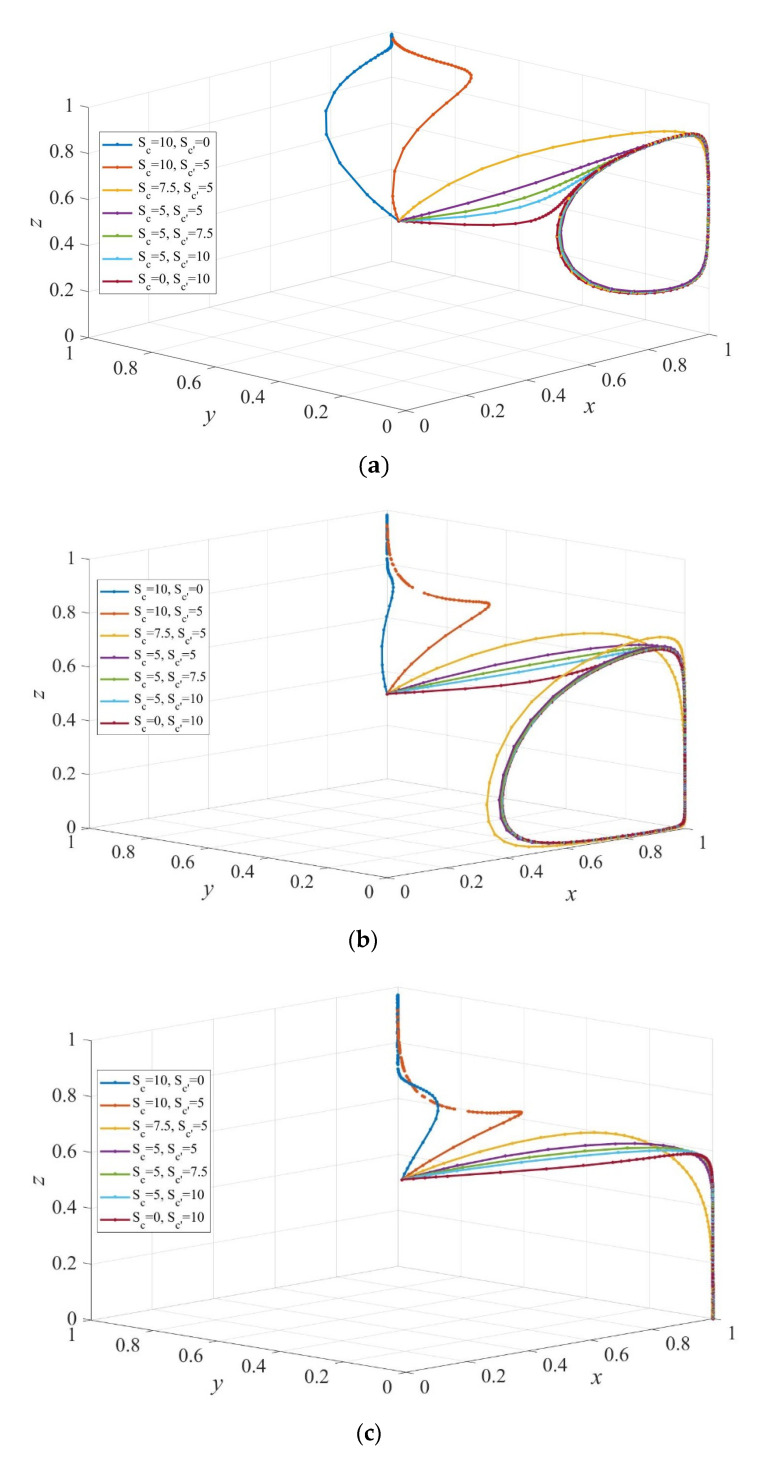
The evolution path of strategies of private sectors, consumers, and governments under different subsidies to consumers and cost savings. ΔC=9 in (**a**), ΔC= 6 in (**b**), and ΔC= 3 in (**c**). Parameters: Q=50, P=18, $S_{1}=6$, $E_{H}=200$, $E_{L}=-50$, β=0.8, $r_{H}=260$, $r_{L}=220,\;R_{H}=390$, $R_{L}=340$, C=80, F=85, α=0.6.

**Figure 9 ijerph-19-02800-f009:**
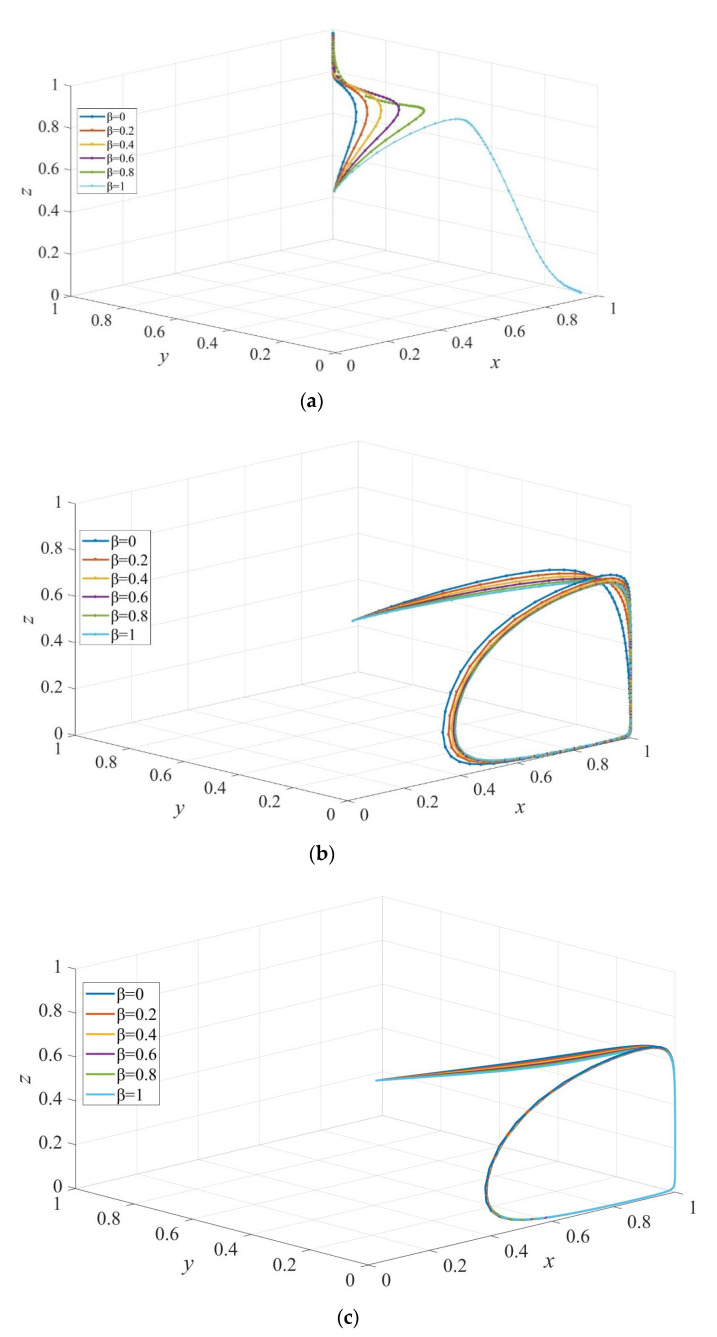
The evolution path of strategies of private sectors, consumers, and governments under different levels of acceptance to elderly care services. $S_{c}=10$, $S_{c^{\prime }}=$ 5 in (**a**), $S_{c}=5$, $S_{c^{\prime }}=$ 5 in (**b**), and $S_{c}=5$, $S_{c^{\prime }}=10$ in (**c**). Parameters: Q=50, P=18, $S_{1}=6$, ΔC=6 $E_{H}=200$, $E_{L}=-50$, β=0.8, $r_{H}=260$, $r_{L}=220,\;R_{H}=390$, $R_{L}=340$, C=80, F=85, α=0.6.

**Figure 10 ijerph-19-02800-f010:**
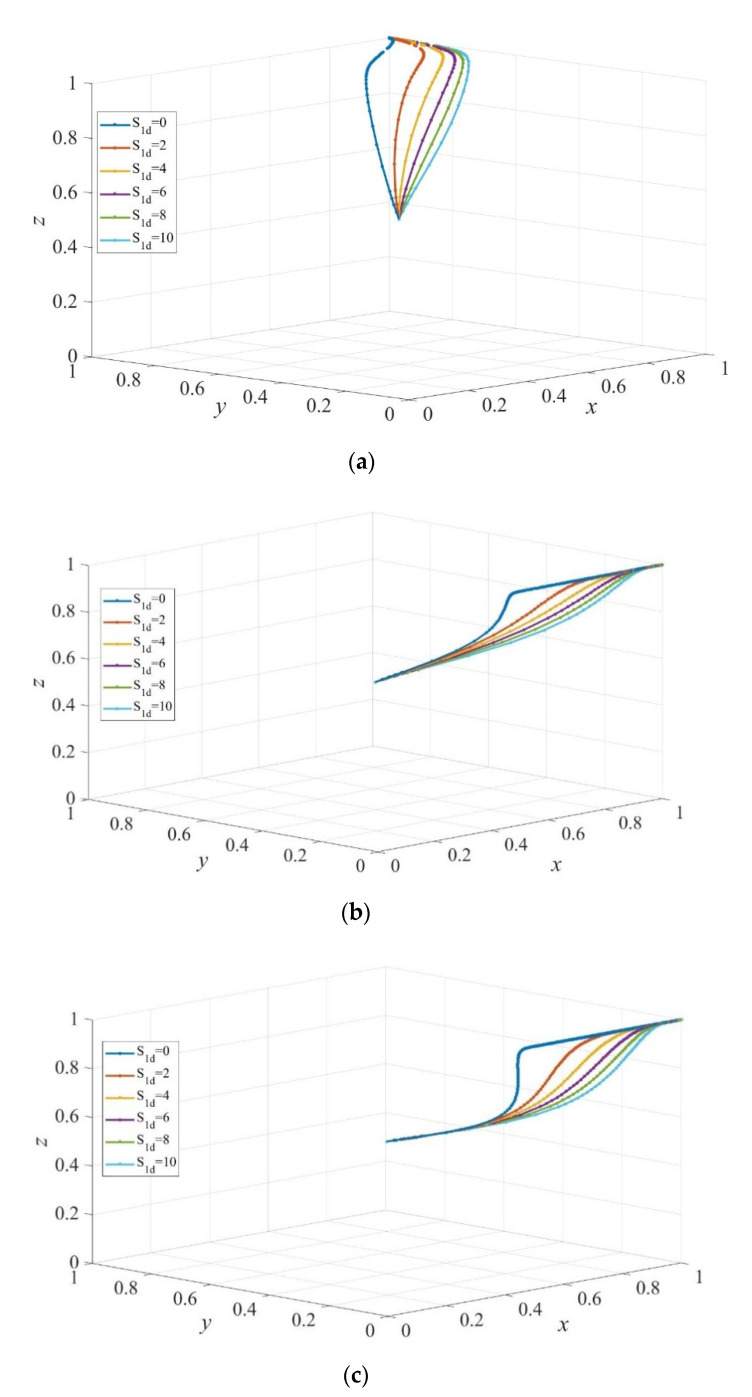
The evolution path of strategies of private sectors, consumers, and governments under different dynamic sizes of subsidy to private sectors. $S_{cd}=10$. $S_{c^{\prime }d}=$ 5 in (**a**), $S_{cd}=5$, $S_{c^{\prime }d}=$ 5 in (**b**), and $S_{cd}=5$, $S_{c^{\prime }d}=10$ in (**c**). Parameters: Q=50, P=18, ΔC=6, $E_{H}=200$, $E_{L}=-50$, β=0.8, $r_{H}=260$, $r_{L}=220,\;R_{H}=390$, $R_{L}=340$, C=80, F=85, α=0.6.

**Figure 11 ijerph-19-02800-f011:**
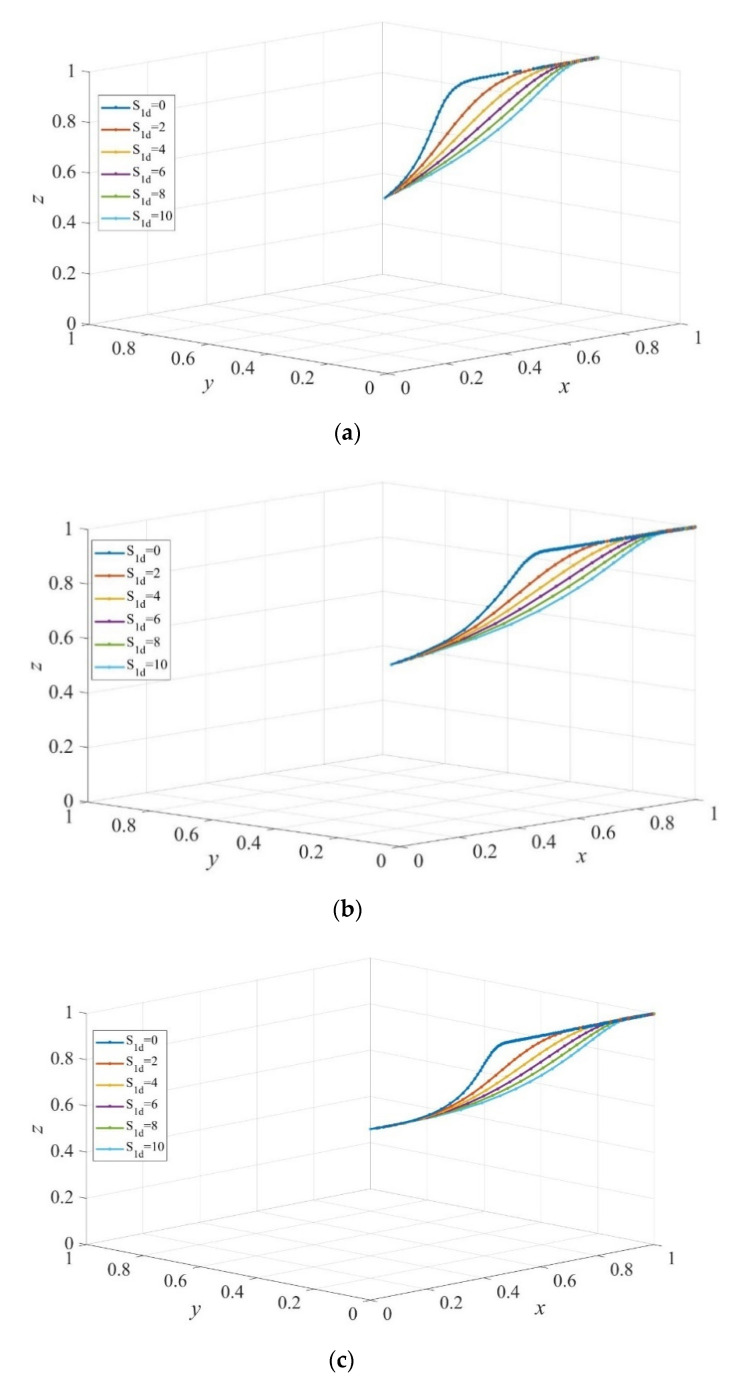
The evolution path of strategies of private sectors, consumers, and governments under different dynamic sizes of subsidy to private sectors and consumers. $S_{cd}=10$, $S_{c^{\prime }d}=$ 5 in (**a**), $S_{cd}=5$, $S_{c^{\prime }d}=$ 5 in (**b**), and $S_{cd}=5$, $S_{c^{\prime }d}=10$ in (**c**). Parameters: Q=50, P=18, ΔC= 6, $E_{H}=200$, $E_{L}=-50$, β=0.8, $r_{H}=260$, $r_{L}=220,\;R_{H}=390$, $R_{L}=340$, C=80, F=85, α=0.6.

**Table 1 ijerph-19-02800-t001:** The payoff matrix.

		Governments			
		*AS (z)*		*NS (1−z)*	
		Consumers		Consumers	
		*IC (y)*	*HC (1−y)*	*IC (y)*	*HC (1−y)*
Private sectors	*HQ* *(x)*	$Q(P-C_{H})+QS_{1}+r_{H}$, $-QP+QS_{c}+E_{H}$, $R_{H}-QS_{1}-QS_{c}-C$	$\alpha Q\gamma (P-C_{H})+QS_{1}+r_{L}$, $-\alpha QP+\alpha QS_{c^{\prime }}+\beta E_{H}$, $R_{L}-QS_{1}-\alpha QS_{c^{\prime }}-C$	$Q(P-C_{H})+QS_{1}+r_{H}$, $-QP+QS_{c}+E_{H}$, $-QS_{1}-QS_{c}$	$\alpha Q\gamma (P-C_{H})+QS_{1}+r_{L}$, $-\alpha QP+\alpha QS_{c^{\prime }}+\beta E_{H}$, $-QS_{1}-\alpha QS_{c^{\prime }}$
	*LQ* *(1-x)*	$Q(P-C_{L})-F$, $-QP+QS_{c}+E_{L}$, $-QS_{c}-C+F$	$\alpha Q\gamma (P-C_{L})-F$, $-\alpha QP+\alpha QS_{c^{\prime }}+\left(1+\beta \right)E_{l}$, $-\alpha QS_{c^{\prime }}-C+F$	$Q(P-C_{L})+QS_{1}$, $-QP+QS_{c}+E_{L}$, $-QS_{1}-QS_{c}$	$\alpha Q\gamma (P-C_{L})+QS_{1}$, $-\alpha QP+\alpha QS_{c^{\prime }}+\left(1+\beta \right)E_{L}$, $-QS_{1}-\alpha QS_{c^{\prime }}$

**Table 2 ijerph-19-02800-t002:** Equilibria points and characteristic values.

Equilibria Points	$\lambda_{1}$	$\lambda_{2}$	$\lambda_{3}$
$E_{1}=\left(0,0,0\right)$	$r_{L}-Q\Delta C$	$-\beta E_{L}-\alpha QS_{c^{\prime }}-\left(1-\alpha \right)QP+QS_{c}$	$\left(F+QS_{1}\right)-C$
$E_{2}=\left(0,0,1\right)$	$r_{L}-Q\Delta C+QS_{1}+F$	$-\beta E_{L}-\alpha QS_{c^{\prime }}-\left(1-\alpha \right)QP+QS_{c}$	$-\left[\left(F+QS_{1}\right)-C\right]$
$E_{3}=\left(0,1,0\right)$	$r_{H}-Q\Delta C$	$-\left[-\beta E_{L}-\alpha QS_{c^{\prime }}-\left(1-\alpha \right)QP+QS_{c}\right]$	$\left(F+QS_{1}\right)-C$
$E_{4}=\left(1,0,0\right)$	$-\left[r_{L}-Q\Delta C\right]$	$\left(1-\beta \right)E_{H}-\alpha QS_{c^{\prime }}-\left(1-\alpha \right)QP+QS_{c}$	$R_{L}-C$
$E_{5}=\left(0,1,1\right)$	$r_{H}-Q\Delta C+QS_{1}+F$	$-\left[-\beta E_{L}-\alpha QS_{c^{\prime }}-\left(1-\alpha \right)QP+QS_{c}\right]$	$-\left(F+QS_{1}\right)+C$
$E_{6}=\left(1,0,1\right)$	$-[r_{L}-Q\Delta C+QS_{1}+F]$	$\left(1-\beta \right)E_{H}-\alpha QS_{c^{\prime }}-\left(1-\alpha \right)QP+QS_{c}$	$-(R_{L}-C)$
$E_{7}=\left(1,1,0\right)$	$-[r_{H}-Q\Delta C]$	$-\left[\left(1-\beta \right)E_{H}-\alpha QS_{c^{\prime }}-\left(1-\alpha \right)QP+QS_{c}\;\right]$	$R_{H}-C$
$E_{8}=\left(1,1,1\right)$	$-\left[r_{H}-Q\Delta C+QS_{1}+F\right]$	$-\left[\left(1-\beta \right)E_{H}-\alpha QS_{c^{\prime }}-\left(1-\alpha \right)QP+QS_{c}\;\right]$	$-\left(R_{H}-C\right)$

## Data Availability

The study did not report any data.

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
