# Peer review of "Optimal Subsidy Support for the Provision of Elderly Care Services in China Based on the Evolutionary Game Analysis"

_ijerph, 2022, doi:10.3390/ijerph19052800_

Round 1

Reviewer 1 Report

Please see the attached report.

Reviewer 2 Report

This paper studies how subsidy support of the government for elderly care services affects decision makings of stake holders (i.e., private sectors and consumers). The whole analysis is based on evolutionary game theory. By modeling the relationships among the government, private sectors and consumers as a game, the authors look for conditions under which high-quality of elderly care services are sustained. Even though the model is not applied to general situations but restricted to China, the paper provides some useful information and knowledge to a wide range of readers. I think, the paper can basically be accepted for publication after the authors have dealt with the following comments I point out.

(1) Since the authors focus on the situations in China, I would suggest including the word “China” or “Chinese” in the title, in order to avoid misunderstandings of readers.

(2) Evolutionary game theory assumes that there are many players that play the same game. In this present paper, three types of players are assumed: private sectors, consumers and the government. Do the authors assume that there is only one government as a player? If so, the decision-making process of the single player cannot be analyzed by standard evolutionary game theory. Please make the basic assumption clear. Do the authors consider situations where there are many governments (local governments)? In this case, the third type of players should be referred to as “governments”, not as “the government”.

(3) Table 1 is hard to see. I would suggest that the authors show payoff matrices for the three types of players separately (for instance, Table 1(a) for the payoff matrix of private sectors, Table 1(b) for consumers and (c) for governments.)

(4) Around line 251 in page 6, please insert more explanations about I_11 and I_12. What the indices 11 and 12 mean?

(5) The authors mention 8 fixed points that correspond to the vertices of the state space (cube). Are not there any internal fixed points? For example, according to Figure 6, it seems that there is an internal fixed point in the center of periodic trajectories in the plane x-z.

Round 2

Reviewer 1 Report

Please see the attached report.
